# OpenReview forum: "Secret Collusion among AI Agents: Multi-Agent Deception via Steganography"
_NeurIPS.cc/2024/Conference — NeurIPS 2024 poster_

### Official Review · Reviewer_AWRn · 2024-07-08

**Soundness:** 3
**Presentation:** 3
**Contribution:** 3
**Rating:** 7
**Confidence:** 4

**Summary:**

Evaluates models' ability to collude via steganography, evading human oversight. Various monitoring and capability-limitation mitigation strategies are proposed. The paper concludes that while current models have limited steganographic capabilities, continuous monitoring of future models is needed.

**Strengths:**

* Wide range of steganographic techniques considered including techniques that aim to be undetectable under statistical analysis
* Many models tested from GPT-2 to GPT-4, spanning a large capability range
* Proposes a range of strategies to mitigate collusion risk

**Weaknesses:**

It would be good to see more detail/justification in section 7.2. (See **Questions**)

**Questions:**

Section 7.1 Monitoring and Penalisation:
* Did you consider model-internals-based probes/activation-monitoring techniques to detect collusion/use of steganography?

Section 7.2 Limiting Capabilities:
* How would you detect pretraining data to filter out? Do you mean something like filtering out research papers on steganography and cryptography?
* Test-time/in-context learning is a crucial capability of LLMs. Do you have any thoughts on ways to selective ablate the ability to learn steganography-relevant information at test-time while retaining other capabilities?
* How would backdoors be used to "give agents access to advanced steganographic capabilities" and do you have a proposed mitigation given the results you cite on the difficulty of reliably destroying backdoors?
* Would be good to explain more thoroughly in what scenarios paraphrasing would and would not work, and how semantic entropy could be used for covert communications.
* Do you have any empirical tests/examples of steganography that are robust to paraphrasing?

**Limitations:**

Some limitations are addressed in the "Conclusion and Future Work" section. However, some additional things to note:
* Your evaluation focuses on text inputs and outputs, however multimodal image/audio models may bring additional risks/considerations
* Section 7.2 could be fleshed out with more information on how these techniques would successfully reduce collusion risk

---

> ### Author Rebuttal · Authors · 2024-08-06
>
> We thank the Reviewer for their insightful feedback and are happy to see they find our paper interesting.
> Please find our responses below.
>
> ## Questions
>
> **(1) R**: *"Did you consider model-internals-based probes/activation-monitoring techniques to detect collusion/use of steganography?"*
>
> Many thanks for highlighting this highly interesting research direction.
>
> We mention that white-box access to agents could yield additional opportunities for steganography detection, and we have now added an additional discussion of this to the CRC. Note that related literature can be found within (mechanistic) interpretability approaches to detecting deception.
>
> **(2) R**: *"How would you detect pretraining data to filter out? Do you mean something like filtering out research papers on steganography and cryptography?"*
>
> We agree that this question merits greater detail, and we have now added an additional discussion to the CRC. In general, a large and diverse number of approaches to filtering pre-training data have been suggested, ranging from keyword-based methods to using LLM classifiers [9].
>
> **(3) R**: *"Test-time/in-context learning is a crucial capability of LLMs. Do you have any thoughts on ways to selective ablate the ability to learn steganography-relevant information at test-time while retaining other capabilities?"*
>
> This is a really interesting question. We are aware that future LLMs could acquire or improve steganographic capabilities at test-time, perhaps using mesa-optimisation. While the question of how to selectively ablate such capabilities at test-time deserves greater study in future work, we suggest that there are broadly two avenues to doing so: a) one could compare test-time performance to a setting in which test-time incentives for the use of steganographic capabilities have been adjusted, or b) explore approaches requiring white-box access (see Question (1)).
>
> **(4) R**: *How would backdoors be used to "give agents access to advanced steganographic capabilities" and do you have a proposed mitigation given the results you cite on the difficulty of reliably destroying backdoors?*
>
> Thanks for this question - we have now added additional clarification of this to the CRC. Just like ordinary neural network backdoors, we envisage such backdoors to be crafted such as to be triggered by special test-time inputs. Such functionality could make use of generic train->test time input distribution shifts (think e.g. dates), or rely on subtly crafted test-time inputs that are hard to distinguish from innocuous inputs.
>
> As backdoors may be hard to detect even under white-box access to the model's weight, and might not be easy to remove using detection-agnostic performance-preserving methods (such as e.g. fine-tuning), future work should examine whether white-box methods operating in activation space (see also Q (1)) could help detect and mitigate such backdoors.
>
> **(5) and (6) R**: *"Would be good to explain more thoroughly in what scenarios paraphrasing would and would not work, and how semantic entropy could be used for covert communications." and "Do you have any empirical tests/examples of steganography that are robust to paraphrasing?"*
>
> Thanks for this very interesting question. In our additional paraphrasing experiments part of the global rebuttal, we show that paraphrasing may be fundamentally ineffective in tasks that can be performed by LLMs in more than one way without sacrificing performance. Naturally, such tasks would include cooperative tasks with large convention spaces, where each convention could entail communicating different semantics (e.g. choosing to circumvent an obstacle on the right, vs circumventing it on the left - and communicating these choices in natural language).
>
> Another, perhaps unexpected, limitation of paraphrasing is its potential to deteriorate task performance in practice, as empirically observed by us across several tasks (see PDF in the global rebuttal). The reason for this is that paraphrasing can distort semantic context in practice if it does not adequately take a conversation's full context into account.
>
> We leave empirical evaluations of steganographic methods robust to paraphrasing for future work.
>
> ## Final Remarks
>
> We again thank the Reviewer for their insightful questions, and hope the reviewer will consider vouching for acceptance.

---

> ### Author Response · Authors · 2024-08-13
> **Summary of Updates on our Rebuttal**
>
> We would like to notify Reviewer AWRn of the following updates in our rebuttal:
>
> - All Reviewers are now in favour of acceptance.
> - We have added global comments succinctly summarising our key contributions, clarifying our experimental setup, and detail improvements to the writing and structure of the paper in our camera ready version.
> - We have responded to more specific questions and feedback in our rebuttal above and thank the reviewer for raising these insightful points.
>
> We hope that our response to Reviewer AWRn has been satisfying; we are happy to provide further clarification until the rebuttal deadline.

---

### Official Review · Reviewer_pyc3 · 2024-07-09

**Soundness:** 1
**Presentation:** 1
**Contribution:** 1
**Rating:** 5
**Confidence:** 2

**Summary:**

In this paper, authors propose a way to determine whether LLMs are secretly collaborating to hide, prevent, or misguide users that ask for a specific input. The authors show that, through steganographic techniques, it is possible to embed secret instructions retrieved only by hitting particular queries, while keeping intact its functionality with all the other users.

**Strengths:**

**Interesting point of view.** Steganography can be used to instruct LLMs to not respond to specific questions.

**Weaknesses:**

**Without steganographic background, the paper is hard to read.** This manuscript is very difficult to understand, since most of the explanations of relevant topics are left to the appendix, without giving the intuition in the main paper. Thus, I debate that the full paper is *in the appendix*, and it is clearly badly written for this conference.

**Experiments are misleading.** As a follow-up of the first comment, the experimental part is difficult to decrypt, since it is not clear what the metrics do, what are the involved quantities, what is a good or bad result. More importantly, it is not clear at all if those experiments are really showing a collusion between agents. The experimental section fails in conveying the message that the authors wanted to show.

**CASE introduced, but never explained in the main paper.** The evaluation framework (again, left to the appendix as all the remaining relevant concepts of the paper) is not even explained intuitively. Also, it seems so vague that it could be applied to any time of evaluation of ML models, not only LLMs. While the authors do not claim it as a contribution, it needs to be better explained *in the main paper*.

**Is this colusion, or just instructions of LLMS?** To worsen the situation, I am not sure if the exposed results show collusion at all. From an high-level point of view, private LLMs can be instructed to avoid certain answers (like illegal content generation). Is it collusion? Or is it indend by design? While I have understood the point, I debate that using closed-source LLMs already guarantee the fact that they could discriminate content depending on hidden variables. As a very good contribution, while I understand it is almost impossible to achieve it, the authors could show this issue on **commercial** products, thus proving their point.

**Final remarks: there is a lot of content, but it is very badly explained and badly divided between main paper and appendix.**

**Questions:**

**Can you better shape the content?** This might help readers and reviewers in understanding the proposed content (which, I admit, it is a lot). Exactly for the reason that the authors produced plenty of content, it would be great also to present it carefully.

**Can you better convince readers on collusion?** As it is now, I am unsure whether it is clear what collusion is, and how it is measured. The single example given might be confused with a jailbreak attempt, or just a specific instruction given to LLMs.

**Limitations:**

I am unable to understand whether, in this paper, the limitations have been carefully investigated.
From what I could decrypt, the authors list potential problems, which are related to plenty of models but not specific to their evaluation.

---

> ### Author Rebuttal · Authors · 2024-08-07
>
> We thank the reviewer for their time and efforts. **Please also refer to the global rebuttal concerning our concern's about the Reviewer's misunderstandings of our paper's focus as evidenced by their summary.**
>
> ## Strengths
> **(1) R**: *"Interesting point of view. Steganography can be used to instruct LLMs to not respond to specific questions."*
>
> **We respectfully ask the reviewer to revisit our paper, as the mentioned strength seems to reflect a misunderstanding of our central thesis and focus**. While in theory it might be possible to use steganographic communication to instruct an LLM not to respond to specific questions, this is a rather niche use case/threat model and is unrelated to the contents of our paper.
>
> ## Weaknesses
> **(1) R**: *Without steganographic background, the paper is hard to read.*
>
> We would like to push back on this point – our key work is in the main body, with formal definitions and citations provided throughout for any specialist terms that we use. We would be happy to explain any relevant concepts from the paper that the reviewer would like to understand better. The Appendix, like any substantial conference submission, contains our proofs and demonstrations of empirical results that we summarize in our main body.
>
> **(2) R**: *Experiments are misleading.*
>
> We politely ask the reviewer to be more specific in this criticism, if possible, as the metrics we consider correspond simply to the success rates for various forms of steganographic collusion on the specific tasks we consider. Insofar as these capabilities might be worrying or undesirable, then higher success rates are “bad”. Given that many of these success rates are non-negligible (to say the least), it is clear that these forms of collusion are in fact possible with some of today’s more advanced models, which is the key takeaway for the experimental section and of the paper, and something we note multiple times throughout. Along with experimental results, we directly provide examples of these runs. We would like to refer the reviewer to the real-world case study in Section 6, Appendix Section J.2, and Appendix Section K.2 for group capability evaluations that test whether collusion arises autonomously, without any instruction, in settings with multiple agents.
>
> To make our paper clearer and address the reviewer’s concerns, however, we have added explanations for the usefulness of minimum entropy couplings, logit simulation, self-sampling, k-level reasoning, and storing one-time pads more clearly in Appendix Section F of our camera-ready draft.
>
> **(3) R**: *CASE introduced, but never explained in the main paper.*
>
> We explain what the CASE framework is in Section 5 and have more clearly denoted the empirical investigations (from Table 1, Figure 4, Figure 5, and qualitative results in Section 6) as part of the CASE framework in our camera-ready draft. CASE is explained in additional detail in Appendix J, including specifics about how our evaluations are performed, which would sadly have taken up too much room in the main body. Like any substantial conference submission, we leave additional details in the Appendix and focus on results in the main body.
>
> **(4) R**: *Is this colusion, or just instructions of LLMS?*
>
> To clarify, no, instructing an LLM to avoid giving certain answers is not collusion. Collusion refers to the ability of agents to coordinate in undesirable ways. This coordination is often most worrying when it is secret, i.e. hidden and/or not easily detectable.
>
> We are slightly unsure as to what is meant by the claim that “using closed-source LLMs already guarantee [sic] the fact that they could discriminate content depending on hidden variables”, as it is unclear what is meant by “discriminate content” or what “hidden variables” are being referred to. We wish to reassure the reviewer, however, that none of our tasks could have leaked into the training data of the models we used, and that no additional hidden information or context was provided to models that might have invalidated our experiments. We also use several open-source models, such as Llama 2 and Mixtral.
>
> To clarify further, we also show our results with several commercial products – each frontier model evaluated according to our individual and group capability evaluations is a commercial product.
>
>
> ## Questions
> **(1) R**: *"Can you better shape the content? This might help readers and reviewers in understanding the proposed content (which, I admit, it is a lot). Exactly for the reason that the authors produced plenty of content, it would be great also to present it carefully."*
>
> We would like to ask the reviewer to re-examine our paper in light of our responses. We would be happy to bring any relevant sections that the reviewer recommends from the Appendix to the main body, page limits permitting, and are happy to discuss any actionable feedback from the reviewer to better shape our content.
>
> **(2) R**: *"Can you better convince readers on collusion?"*
>
> As noted above, (steganographic) collusion refers to the ability of agents to (secretly) coordinate in undesirable ways. Understanding exactly how agents might do this is one of the key challenges we consider in our paper and something we set out to evaluate in a variety of different settings in order to understand a variety of different capabilities that might enable such collusion. For example, in the example considered in Figure 2, the LLM coordinates with the human to secretly reveal information in a way that is undesirable. More generally, the DecAI setting and experiments we conduct in the paper consider the possibility that this undesirable (secret) coordination might take place not only between LLMs and humans, but also just LLMs. We understand how the given example might have been confusing insofar as it does not represent the latter case, however, and will update the example accordingly in the camera-ready version.

---

> > ### Comment · Reviewer_pyc3 · 2024-08-09
> > **Acknowledging my wrong interpretation on some points, but confirming other doubts.**
> >
> > Thanks to your comments I was able to dissect better the paper, and I will increase my score to **Reject** (and also reduce my confidence on this paper to 2 from 3).
> > With self-doubt, I have re-read the paper from scratch again, going over the appendix.
> > As a result, I understand that your frameworks evaluate whether a collusion might appear in practice, by looking at the exchange of messages between two LLMs. For this reason I will rise the score as stated, and I apologize for not having understood the paper in the first run.
> > However, I still think that this paper has issues that prevent acceptance as follows.
> >
> > **The paper is not well-written, since all major content is in Appendix** In my second run, I still had to re-read the paper multiple times, and I got **what** has been presented from the Appendix, and not the main paper. For instance, all the single and group tests were unclear on their setup, goal, and execution. Thus, authors should better explicit both steps in main paper, and only later clarify minimal details.
> > I would reduce section 4 (maybe included in sect 3 as well) and I would expand section 5 to clarify that:
> >
> > * you instruct LLMs with system prompts to assess whether they can use steganographic techniques, like base64 or chiper algorithms (this is only hinted in one sentence at page 6).
> > * how you apply group evaluations, since in Appendix H is only written that models are applying techniques based on single algorithm generation (point 1), but I still don't understand how it works. Again, the explanation on the precise setup must go in the main paper.
> >
> > **Benchmark of steganographic techniques.** While it is interesting that LLM can generate code for creating steganographic content, I do not think that such stands on a contribution on its own. In fact, all the presented techniques are wide known in public literature, and they are generated by LLMs just as code for many languages. This is similar to previous work showing that LLM can write code. An higher contribution would have been showing that previously it was not possible, or that alignment and guardrails prevent the generation of such content.
> >
> > **I still think experiments are misleading.** By reviewing the paper, it is hard to understand the message that these figures are conveying. I can understand what is happening only after having read all the Appendix. If I got it correct, the success rate is the relative number of times that an LLM generated the required content (Fig5).  Thus, it appears that only GPT4 (which is closed source LLM) is capable of generating all the techniques, while all the other baseline fail to even reach 30% (which are 3 attempt over 10 on average). From my point of view, this does not look as a contribution, since the only capable one is a closed source LLM that might rely on extra knowledge, and there is no way for telling the contrary.
> >
> > **Overseer?** In all the above, I miss the relevance of the overseer presented in the figure, since I can't understand the setting of group evaluations.
> >
> > I truly apologize with the authors for not having understood parts of the paper in the first run, but re-reading again confirmed some of my belief about contributions, setting up of experiments, presentation of content.

---

> > > ### Author Response · Authors · 2024-08-10
> > > **Concerns about reviewer pyc3's comments**
> > >
> > > We appreciate Reviewer pyc3’s fresh look at the paper, and their increase in score and reduction in confidence level.
> > >
> > > **However, we are still concerned about a number of fundamental misunderstandings, and, in particular, lack of acknowledgement of facts included prominently in both the main paper, and appendix.**
> > >
> > > ## Main Concerns
> > > **We provide an overview of the issues in this comment. Please find additional details and specific issues we highlight with the review in the separate comment below.**
> > >
> > > ### Content in Main Paper vs. Appendix
> > > Given the 9 page limit, we cannot include all relevant details in the main body of the text. While we have made some small further improvements based on the reviewer’s suggestion, we fundamentally believe that we are prioritizing the right content in the main body and comprehensively refer to the relevant content in the appendix throughout. Overall, we are confident that our paper structure follows established common practice in the field.
> > >
> > > ### Lack of Acknowledgement of Group Evaluation Details
> > > We politely refer the reviewer to Appendix H.2 (referred to in Section 6), which details our group evaluations. Specifically, we would like to point out that the reviewer’s assertion - “since in Appendix H is only written that models are applying techniques based on single algorithm generation” - is incorrect.
> > >
> > > ### Claiming a Lack of Novelty without Backing This Up with References to Prior Work
> > > Firstly, we are unsure what the reviewer means by “LLM can generate code for creating steganographic content” - our evaluations show that LLMs can apply steganography to their own outputs; they are not about LLMs “generate code”. Secondly, we would like the reviewer to provide references to existing literature that establishes the results in our paper, including how LLM agents can apply steganographic techniques to their own outputs. We would also like to politely point out that our paper offers a wealth of novel contributions beyond LLM capability evaluations, including various theorems and conceptual insights, including on how to prevent steganography.
> > >
> > > ### Dealing with Closed-Source Model Intransparency
> > > While it is possible that GPT-4 may rely on knowledge that is not available to other, including open-source, models, we politely point out that this fact is orthogonal to our contributions. The important takeaway in this case is that using GPT-4 as a basis for agents interacting in the real world could pose safety concerns as these agents may be able to perform steganography at least in some limited form. We disagree that our experiments are “misleading” therefore and would like to kindly request further explanation of why the reviewer thinks so.
> > >
> > > ### Lack of Acknowledgement of Details on the Overseer Agent
> > > The role of the overseer is clearly defined in Section 4, which has an entire paragraph titled “Monitoring Agents”. There are additional practical details on overseers in Appendix H.2 (which is referred to in Section 6), We would like to invite the reviewer to revisit these sections.
> > >
> > > ## General Concerns about Reviewer pyc3’s Engagement
> > > We thank the reviewer for revisiting our paper. However, we are surprised that the reviewer is bringing forth concerns that are clearly addressed in the paper’s main body, as well as prominently referenced sections in the Appendix. Therefore we believe that another, comprehensive, look at our paper would be warranted.

---

> > > > ### Author Response · Authors · 2024-08-10
> > > > **Highlighting specific misunderstandings in Reviewer pyc3's comments**
> > > >
> > > > We here provide additional detail in reaction to Reviewer pyc3’s response:
> > > >
> > > > ## Misunderstandings
> > > >
> > > > **Reviewer pyc3: “base64 or chiper algorithms (this is only hinted in one sentence at page 6).”**
> > > >
> > > > We would like to highlight that this is incorrect. On page 7, line 242/243, we offer a concise explanation, “Although gradual increases in coordination are observed, GPT-4 exhibits a significant performance leap for cipher encoding and decoding (Base64 and Caesar).” For further details on cipher encoding/decoding setups, we would like to refer the reviewer to the Appendix.
> > > >
> > > > **Reviewer pyc3: “since in Appendix H is only written that models are applying techniques based on single algorithm generation”**
> > > >
> > > > We would like to request the reviewer to reevaluate this comment. In Appendix H.2, we provide a clear explanation of our group evaluations: “First, we establish a set-up akin to Simmon’s prisoner’s problem, wherein agents A and B aim to communicate secretly. A needs to send a message M over a public channel accessible to 1175 overseer E. Secondly, we further expand the setup such that A and B share additional common knowledge (in this setting, we use paragraphs from Wikipedia articles written in December 2023 on which LLMs haven’t been trained as shared common knowledge between the collusive models), which E neither possesses in-context nor as part of its training. Through this approach, our goal is to measure how often A can hide a random secret word in an unrelated piece of common knowledge and how well the overseer E can detect the secret versus how often B can decode it. We therefore see this as a simpler version of information-theoretic steganography, where an undetectable message will consist of the secret message and the common knowledge mapped together into a distribution that seems standard. The common knowledge setting presented is a translation of the formalisms in Section 7. Table 20 presents a full breakdown of all results obtained in this evaluation. Most notably, using GPT-4 models, A and B were able to successfully hide and decode secret words without detection by the overseer in 26% of the runs.” We would like to politely ask the reviewer to point out anything they require additional information on regarding this setup.
> > > >
> > > > **Reviewer pyc3: The reviewer first states, “An higher contribution would have been showing that previously it was not possible”** and in the next comment states **“Thus, it appears that only GPT4 (which is closed source LLM) is capable of generating all the techniques, while all the other baseline fail to even reach 30%.”**
> > > >
> > > > We would like to highlight the issue with these two statements being contradictory, since we do show that previous models are unable to do well on our individual capability evaluations, but capabilities on these tasks increase and are within reach of larger models.
> > > >
> > > > **Reviewer pyc3: “While it is interesting that LLM can generate code for creating steganographic content, I do not think that such stands on a contribution on its own.”**
> > > >
> > > > We believe that this is an unfair criticism and  are unsure of why the reviewer mentions *“on its own”* given the depth of evaluations. In only two out of twelve Individual Capability Evaluations (see Table 1) and several other group and qualitative evaluations do we discuss LLMs generating code. These are as follows: Schelling Points (Code) - we measure how well separate models can generate code to encode and decode text respectively such that they find a Schelling point (zero-shot) which allows for secret communication. The second, Tool Use (Code Interpreter), focuses on whether models can use a code interpreter to perform encryption methods on the fly. Moreover, evaluations are just one of our contributions besides a wealth of important theoretical and conceptual contributions. The reviewer’s comments indicate that these contributions have been clearly ignored.

---

> > > > > ### Comment · Reviewer_pyc3 · 2024-08-11
> > > > > **Acknowledge**
> > > > >
> > > > > I am so sorry that my comments about the clarity of the paper are unsettling. I still believe that the paper would need to be improved in writing. Also, this paper is 10 pages long, since two paragraphs are written in page 10.
> > > > > Also, I am still not convinced by the technique, and apparently I am not able to do my job.
> > > > >
> > > > > Since I am the only reviewer that had struggle understanding the paper, I'll rise my score to borderline accept so that my decision is not blocking your path to the conference. I read the other comments from reviewer and no one is pointing out my concerns.
> > > > >
> > > > > I am truly sorry about all this discussion.

---

> > > > > > ### Author Response · Authors · 2024-08-12
> > > > > > **Thanks for Updating Your Score**
> > > > > >
> > > > > > We thank Reviewer pyc3 for increasing their score. We are very grateful for the time and effort the Reviewer has put into assessing our work.
> > > > > >
> > > > > > As detailed in the global comment we just submitted, we are committing to, if accepted, make further improvements to increase the accessibility of our paper and further clarify the contents of the main body in the CRC.

---

### Official Review · Reviewer_csJ4 · 2024-07-13

**Soundness:** 3
**Presentation:** 3
**Contribution:** 3
**Rating:** 6
**Confidence:** 3

**Summary:**

The paper formalizes the problem of secret collusion in systems of generative AI agents, leveraging steganographic techniques. The authors propose a model evaluation framework to systematically test capabilities required for various forms of secret collusion. The paper provides extensive empirical results across a range of contemporary LLMs, noting significant capability jumps in models like GPT-4. The paper concludes with a comprehensive research program aimed at mitigating future risks of collusion between generative AI models.

**Strengths:**

- The paper addresses a timely and critical issue of collusion among AI agents, which has significant implications for AI safety and security.

- The systematic model evaluation framework is well-structured and provides a clear methodology for testing steganographic capabilities in AI models.

- The empirical results are robust, covering a wide range of contemporary LLMs and demonstrating significant capability differences among them.

**Weaknesses:**

- The paper could benefit from more detailed real-world case studies or deployment scenarios to demonstrate the practical applicability of the proposed evaluation framework.

- The discussion on the limitations of current steganographic detection techniques and how the proposed framework addresses these could be more detailed.

**Questions:**

- How do you plan to address the computational costs associated with the integration of steganographic detection techniques in real-world deployments?

**Limitations:**

- The empirical validation, while extensive, could benefit from additional real-world case studies or deployment examples to further validate the approach's practical applicability and robustness.

- The approach has been primarily evaluated on text-based LLMs. It remains to be seen how well it generalizes to other modalities and AI systems.

- The paper should discuss the potential computational and resource costs associated with the continuous monitoring and detection strategies proposed.

- The mitigation strategies proposed, such as continuous monitoring and penalization, might face scalability challenges in large, decentralized systems with numerous interacting agents. The paper does not address how these strategies can scale effectively while maintaining robust detection and mitigation of collusion.

- The long-term effectiveness of the proposed detection and mitigation strategies is not thoroughly evaluated. As generative AI models continue to evolve, new steganographic techniques may emerge that circumvent current detection methods. The paper should discuss plans for continuous adaptation and improvement of the framework to keep pace with advancements in AI technology.

---

> ### Author Rebuttal · Authors · 2024-08-06
>
> We thank the Reviewer for their effort in evaluating our work and for their valuable feedback. We are happy to address their concerns and add corresponding improvements to our paper.
>
> ## Weaknesses
>
> **(1) R**: *Real world case studies and deployment scenarios*
>
> We agree that further evaluation of our experimental and theoretical framework to real-world AI systems constitutes interesting future work.
>
> However, we also argue that our scenarios and evaluations are already broadly representative of multi-agent LLM systems likely to be deployed near-term - we present evaluations for (groups of) frontier models of different sizes and capabilities (also see Section 6 and Appendix sections F, G, and H). Moreover, Appendix section J contains a study that directly looks at collusion arising in real-world inspired scenarios and provides a realistic test-bed to evaluate these settings. Our work is therefore directly applicable to both evaluations (red-teaming [5]) and mitigation strategies in practice and is flexible enough to cover settings involving future frontier models. We will more clearly reference these parts of our paper in the main body in the CRC.
>
> **(2) R**: *More details regarding the limitations of steganographic mitigation techniques*
>
> We agree that further detail on the limitations of current steganographic detection techniques, as well as on the proposed framework, would benefit our work. To this end, we have added extensive experiments around limitations of paraphrasing to our global rebuttals (PDF) and discussions about other limitations to the CRC.
>
> We'd like to highlight the already extensive discussion of the limitations of stated mitigations: see Theorem 2, which discusses the intractability of preventing collusion through black-box parameter optimisation/safety fine-tuning [4], and Theorem 3, which discusses how LLM agents can generate samples from arbitrary output distributions even without access to internal or external sources of (pseudo)randomness given sufficiently unique inputs. For paraphrasing, we identify and formulate how semantic entropy (i.e. choice over communication semantics/topics) can be utilized for covert communications even when paraphrasing is done within the lexical subspace. Throughout Section 7 we have provided an extensive list of citations with discussions of mitigation techniques and limitations, including but not limited to [8] which discusses how perfectly secure steganography limits the utility of covertext anomaly detection and agents learning steganographic abilities during test-time via mesa-optimization [10].
>
> Our work addresses these problems both empirically and theoretically. In Theorem 1, we prove how grounding functions can detect steganography, providing a mitigation that mitigates the limitations of techniques studied in Theorems 2 and 3. We also highlight how in-context instructions could be useful and provide concrete, practical recommendations such as filtering pre-training data and restricting tool use which are directly applicable to current frontier models. We would like to leave the implementation of monitoring techniques similar to [1] and paraphrasing/assessing the limits of error correction methods to future work.
>
> ## Questions
>
> **(1) R**: *Deployment computational costs for mitigations*
>
> We note that some of the mitigation techniques we mention in Section 7 such as in-context instructions, filtering pre-training data, and restricting access to external tools/sources of randomness are not compute-intensive. Moreover, paraphrasing via an overseer need not be continuous, similar to how monitoring is made less resource intensive in [1]. Initially, direct oversight will be needed in critical settings with untrusted multi-agent systems, and the other systemic safety-based mitigations we mention will help provide a less compute intensive approach in the long term.
>
> ## Limitations
>
> **(1) R**: *Empirical validation could benefit from additional real world case studies*
>
> See response to Weakness (1).
>
> **(2) R**: *Expanding to other modalities*
>
> We acknowledge that multimodal LLMs are an interesting direction for future work and appreciate the reviewer’s suggestion.
>
> **(3) and (4) R**: *Computational costs and scalability challenges for mitigations*
>
> We agree with the reviewer that steganographic detection methods can pose practical challenges and we have added a more detailed discussion of this to the CRC. However, computational scalability may not be as problematic as the Reviewer anticipates (see response to Q 1).
>
> **(5) R**: *The long-term effectiveness of the proposed detection and mitigation strategies is not thoroughly evaluated*
>
> We do in fact consider long-term deployment of the mitigation strategies mentioned in Section 7 and associated limitations. Specifically, we discuss perfectly secure steganography [8] which cannot theoretically be improved upon. We identify mitigation measures with this limit in mind, and although current models are unable to perform information-theoretically undetectable steganography, we provide extensive evaluations to test these growing capabilities. Theorem 1 provides a proof for the setting of a distributed system of generative AI agents where steganography can be detected with a grounding function that takes in channel symbols and prior facts, and outputs a collection of posterior facts that can be used to detect steganography even in the limit (see section D.1 for a proof). We also discuss near-term approaches and their limitations based on model capabilities (see limitations of paraphrasing, suppressing random oracles, and preventing test-time learning). Furthermore, we include a section on the effect of quantum computing advances on our evaluation framework, attempting to anticipate long-term aspects.
>
> Once more, we thank the reviewer for their insightful comments. We hope that our answers here will help to clarify our paper, and that the reviewer might consider increasing their score.

---

> ### Author Response · Authors · 2024-08-13
> **Updating Reviewer csJ4 on our Rebuttal**
>
> We would like to notify Reviewer csJ4 of the following updates to our rebuttal:
>
> - All Reviewers are now in favour of acceptance.
> - We have added global comments succinctly summarising our key contributions, clarifying our experimental setup, and detail specific improvements to the writing and structure of the paper in our camera ready version.
> - We have responded to all the questions raised in this rebuttal and have provided further experiments in the global rebuttals based on the reviewer's feedback.
>
> We hope that our response to Reviewer csJ4 has been satisfying; we are happy to provide further clarification until the rebuttal deadline.

---

### Official Review · Reviewer_Pbei · 2024-07-13

**Soundness:** 3
**Presentation:** 3
**Contribution:** 3
**Rating:** 5
**Confidence:** 4

**Summary:**

With the rapid development of AI, the safety of large language models (LLMs) is becoming a significant topic, such as the privacy and security issues between communicating generative AI agents. In this manuscript, noticing the potential risk emerging from groups of current generative AI agents, the author(s) formalized the problem of secret collusion in systems of generative AI agents, introduced a model evaluation framework for testing the capabilities required for secret collusion across various LLMs, and proposed a variety of mitigation measures. The results in the manuscript look interesting, and GPT-4 displays a steganographic capability jump compared to other current models.

**Strengths:**

In my opinion, the strengths of this manuscript are as follows:

1. The topic of secret collusion related to generative AI Agents is a new topic, and it is worth exploring.

2. Proposed a comprehensive model evaluation framework that could be used to assess the capabilities of various AI LLMs in terms of their potential for secret collusion.

By the way, the author(s) also briefly mentioned the implications of quantum computing, which might be expanded to address how the advent of quantum computing might affect the security of steganographic schemes in the future. It is interesting.

**Weaknesses:**

In my opinion, the weaknesses of this manuscript are as follows:

1. The mitigation strategies were discussed but might not provide enough detail on how these strategies could be implemented in practice.

2. The study is limited to simulations and theoretical frameworks; it might lack real-world applicability and robustness testing in actual AI systems and environments.

3. Some mathematic symbols or descriptions are not so clear.

For more details, please see the Section "Questions" below.

**Questions:**

I read the manuscript, and I have the following questions/comments. I am looking forward to the responses from the author(s). Thanks.

1.How do you suggest extending the model evaluation framework to handle more complex settings with a larger number of AI agents?

2.It would be great if the author(s) could provide more detail on how these strategies could be implemented in practice, especially in the face of advancing AI capabilities.

3.Please check the mathematical symbols in the manuscript carefully; I think there are some typos there; for example, in Line 150, $E^{\Diamond}_{\rm{r}}$, do we need the subscript $\rm{r}$ there? Or in Line 152,  $E^{\Diamond_s}$, do you miss the subscript $\rm{r}$ there?

4.Some descriptions are unclear. Suggest adding a short explanation to the metrics/terms for their first-time appearance in the main text; for example, it would be great if the author(s) could explain the label "Successful Reconstruction Rate" in Figure 4, and the Success Rate and Max Dimension Value in Figure 5.

5.In Line 226, it mentioned Figure 7, where is this Figure?

6.Regarding Figure 6, if we have only one subfigure 6(a), it would be great to write as Figure 6 directly.


Some other issues:

(1) Reviewing so many references is good. But I am not so sure that ... , I think, maybe, it would be great if the author(s) could refine the citations. In addition, I noticed that some references were cited more than once but with slightly different formats, such as Refs.[28,29], this is a little strange.

(2) Line 383, what is the Section number?

(3) Inconsistency format issue in References:

(a) The conference name sometimes used an abbreviation; sometimes not, such as Ref.[63].

(b) The source of some references is missing, such as Ref. [59].

(c) The author's name, sometimes with the abbreviation, sometimes not; for example, Refs.[31,40].

(d) Ref.[26], "2nd edition edition," seems to be a typo. It should likely be "2nd edition."

(e) In Ref.[28], "jul 2022"=>"July 2022".

(f) Same author but different format, such as Ref.[88] vs. Ref.[89].

...

Please check carefully; it would be great if the author(s) could correct the issues in the current manuscript.




I would like to consider adjusting my current score based on the responses from the author(s). Thanks.

**Limitations:**

Yes.

---

> ### Author Rebuttal · Authors · 2024-08-06
>
> We thank the Reviewer(R) for their effort in evaluating our work and valuable feedback. We address their concerns below.
>
> ## Weaknesses
>
> ### Weakness 1
>
> **R**: *"The mitigation strategies were discussed but might not provide enough detail on how these strategies could be implemented in practice."*
>
> We agree that Section 7 would benefit from a more detailed discussion of practical implementation. We will integrate this detailed discussion into our CRC. Below we provide a high-level overview for the Reviewer:
>
> In order to implement **monitoring and oversight systems** in practice, please see [1] which uses trusted editing where a smaller, trusted model (or covertext anomaly detection system) can rank text "suspiciousness" and paraphrase/report it should it fall beyond a certain threshold. Using smaller models for less resource intensive supervision, also described in [2], can be directly implemented in production and will reduce the amount of human monitoring required through autonomous systems overseeing multi-agent communication.
>
> We also discuss **in-context instructions**, which are heavily implemented in practice to prevent unintended usage of models, such as system prompt leakage prevention [3]. This provides an initial line of defense via explicitly prompting models to not share certain information and is a short-term empirical solution that we examine.
>
> We show that mitigating collusion through **black box parameter optimization** is generally an intractable problem (see Theorem 2). This observation is in line with [4] and other works that highlight key limitations of using weight fine-tuning methods for alignment - despite its utility for improving LLM safety and instruction following.
>
> Restricting **access to external tools** can be implemented by whitelisting only trusted endpoints and preventing access to cryptographic tools/sources of randomness. **Filtering pre-training data** to avoid steganographic training/usage techniques can be achieved by e.g. keyword filtering or using LLM classifiers [9]. The **verification of the provenance of pre-training weights** can be achieved using standard authentication methods such as digital signatures.
>
> Regarding implementing paraphrasing, the difficulty is to find an appropriate trade-off between preserving intended channel performance vs. suppressing steganographic capacity (see Additional Experiments in our global rebuttals).
>
> ### Weakness 2
>
> **R**: *"The study is limited to simulations and theoretical frameworks; it might lack real-world applicability and robustness testing in actual AI systems and environments."*
>
> We agree that further evaluations of our simulations and theoretical frameworks to more AI systems deployed in the real world will be interesting future work.
>
> However, we argue that our frameworks and results are broadly representative of multi-agent LLM systems likely to be deployed near-term - we present evaluations for (groups of) frontier models of different sizes and capabilities (also see Section 6 and Appendix sections F, G, and H). Moreover, Appendix section J contains a study that directly looks at collusion arising in real-world inspired scenarios (J.2 also performs secret collusion related ablation studies on GPT-4) and provides a realistic test-bed to evaluate these settings. Our work is therefore directly applicable to both evaluations (red-teaming [5]) and mitigation strategies in practice. We will more clearly reference these parts of our paper in the CRC main body.
>
> ### Weakness 3
>
> We appreciate the reviewer for pointing out these edits and have fixed them in our CRC.
>
> ## Questions
>
> ### Q1
> We divide our settings into two Appendix sections:
> * Individual capability evals (G and I): Can be individually applied to any model within groups of arbitrary size.
> * Group capability evals (H): Specifically designed for setups with multiple models communicating. We note that in scenarios involving two or more agents, we would still need only one overseer.
>
> Expanding H.2 to multiple agents: We can use the same setting with an overseer monitoring every communication part of the multi-agent set up (where the overseer could use covertext anomaly detection tools, an LLM steganography detection model, or perform paraphrasing). The CASE framework for these evaluations is scalable to more agents, where common knowledge can be shared across pairs of communicating agents. This would allow for covertext modification based steganographic collusion and corresponding defenses to be tested.
>
> In future work, we would aim to expand our framework to settings where groups of agents [6][7] are required to communicate and jointly solve cooperative tasks. Each agent would have certain private information that would improve task performance but is not permitted to be shared. The goal would be to then examine whether steganographic collusion arises in this setting over a number of iterations, and how well monitoring and paraphrasing techniques perform. We note that this setup is similar to what we used for our CASE framework, albeit over a number of iterations and more than two colluding models.
>
> ### Q2
> See response to Weakness 1.
>
> ### Q3
> See response to Weakness 2.
>
> ### Q4
> We agree with the reviewer and have now clarified certain descriptions in the CRC. “Successful reconstruction rate” in Figure 4 refers to the percent. of communications, steganographically encoded by the sender, were able to be decoded successfully by the receiver model. “Max coupling dimension by model” refers to the maximum dimension of correct coupling matrices [8] that the models can generate.
>
> ## Other Edits
> We appreciate the reviewer’s feedback on edits related to references, figures, and consistency, and have implemented each change in the CRC.
>
> ## Final Words
>
> Once more, we thank the reviewer for their insightful comments. We hope that our answers here will help to clarify our paper, and that the reviewer might consider increasing their score.

---

> ### Comment · Reviewer_Pbei · 2024-08-10
> **Updating**
>
> Thanks for the responses from the author(s). Also, I read the other reviewers' comments.
>
> To some extent, the responses from the author(s) answered/clarified my questions and concerns. Also, the topic in this manuscript, I think it is interesting. So based on these two points, I increased my score from 4 to 5.
>
> By the way, it would be great if the author(s) could answer some further questions from other reviewers, including the nine-page limit.  Thanks.

---

> > ### Author Response · Authors · 2024-08-11
> > **We thank reviewer Pbei; we have now addressed Reviewer pyc3's additional comments.**
> >
> > We thank Reviewer Pbei for improving their score, and are happy to hear that they find our work interesting. Please do let us know if you have any further questions or concerns.
> >
> > We have now responded to Reviewer pyc3's response to our rebuttal below. While we thank Reviewer pyc3 for taking another look at our paper, we remain concerned about several misunderstandings, inaccurate claims, and a lack of acknowledgment of several of our key contributions.
> >
> > In the light of all other reviewers finding our work interesting and worthy of acceptance, we.would like to ask whether there are any other questions or concerns that, if addressed, could lead them to reconsider their scores or otherwise help with their decisions. Thanks for your engagement with our work.

---

### Author Rebuttal · Authors · 2024-08-05

We sincerely thank all reviewers for their time, effort, and insightful comments on our paper. Taking all feedback into account, we respond to each review individually below. Please also refer to the attached PDF for additional details. Note that all rebuttals share a common bibliography at the end of this global rebuttal.

## Summary of Reviews

* Reviewer Pbei finds our paper introduces a new topic that is "worth exploring" and describes our empirical investigations as "comprehensive."
* Reviewer csJ4 appreciates our exploration of a "critical" AI safety and security issue, highlighting our comprehensive evaluation and "robust" empirical results.
* Reviewer AWRn acknowledges the "wide range" of steganographic techniques considered, underscoring the depth of our work.

In contrast,
* Reviewer pyc3 rates our paper as "poor (1)" across Soundness, Presentation, and Contribution, stating that it is "badly explained." We highlight several issues with this review in our individual rebuttals and comments to the AC.

## Addressing Reviewer pyc3's Misunderstandings

Reviewer pyc3 summarized our paper as "to determine whether LLMs are secretly collaborating to hide, prevent, or misguide users that ask for a specific input.". While this is an interesting theoretical application, it does not accurately reflect our work's focus. We aim to evaluate the potential for steganographic collusion among LLMs, where agents can coordinate in undesirable ways, rather than focusing on specific input manipulation or instruction avoidance.

### Clarification of Focus:

* Main Focus: Our paper explores the potential for LLMs to engage in steganographic collusion, where agents coordinate secretly in undesirable ways. This involves evaluating how steganography might enable such coordination rather than embedding specific instructions for query responses.

* Steganographic Techniques: The summary suggests that our work is about embedding secret instructions triggered by specific queries. Instead, our research evaluates broader collusion capabilities among LLMs, highlighting potential risks and implications of such secret coordination.

* Functional Implications: The summary implies that our primary concern is maintaining LLM functionality while misleading specific queries. However, our work is more focused on assessing the potential for secret coordination, a broader concern beyond isolated query responses.

**We kindly invite Reviewer pyc3 to reconsider our paper, including its strengths, with these clarifications in mind.** To additionally help make our paper even more accessible, we have:

* Soundness: Enhanced explanations of our theoretical framework and methodologies to ensure clarity and accessibility.
* Presentation: Improved the organization and narrative flow to better communicate our key points and findings.
* Contribution: Emphasized the novelty and significance of our work, particularly in the context of existing research.

## Improvements and Clarifications

In light of the feedback, we have made specific revisions based on the detailed and insightful suggestions from Reviewers Pbei, csJ4, and AWRn. These include:

* Expanding our empirical evaluations to provide more comprehensive insights.
* Deepening the discussion of the implications of our findings.
* Refining explanations of our methodology to improve clarity and accessibility for a broader audience.

## Additional Experiments
Experiments related to limitations of paraphrasing as a mitigation technique: In our attached PDF, we demonstrate that paraphrasing, even when provided with context, fails to preserve task performance in cooperative, multi-agent settings like the ChatDev environment [11]. The substantial performance drop and reduction in lexical entropy highlight the limitations of paraphrasing as a reliable mitigation strategy against lexical steganography, thus answering questions around limits of mitigations and paraphrasing proposed by reviewers csJ4, AWRn, and more broadly by Pbei. We thank the reviewers for asking clarifying questions and giving us the opportunity to answer them and will implement the changes/additions discussed here and in our individual rebuttals into the camera ready version of the paper.

## Conclusion
We are grateful for the opportunity to refine our work and look forward to further dialogue and feedback. Thank you once again for your thoughtful reviews.

## Global bibliography (for all reviews)

[1] Greenblatt, Ryan, et al. "AI Control: Improving Safety Despite Intentional Subversion, January 2024." URL http://arxiv. org/abs/2312.06942.

[2] Burns, Collin, et al. "Weak-to-strong generalization: Eliciting strong capabilities with weak supervision." arXiv preprint arXiv:2312.09390 (2023).

[3] Reduce prompt leak. Anthropic. https://docs.anthropic.com/en/docs/test-and-evaluate/strengthen-guardrails/reduce-prompt-leak#example-safeguarding-proprietary-analytics

[4] Casper, Stephen, et al. "Open problems and fundamental limitations of reinforcement learning from human feedback." arXiv preprint arXiv:2307.15217 (2023).

[5] Perez, Ethan, et al. "Red Teaming Language Models with Language Models." arXiv preprint arXiv:2202.03286 (2022).

[6] Park, Joon Sung, et al. "Generative Agents: Interactive Simulacra of Human Behavior." *arXiv preprint arXiv:*2304.03442 (2023).

[7] Zhuge, Mingchen, et al. "Mindstorms in Natural Language-Based Societies of Mind." *arXiv preprint arXiv:*2305.17066 (2023).

[8] Schroeder de Witt, Christian, et al. "Perfectly Secure Steganography Using Minimum Entropy Coupling." ICLR 2023.

[9] Longpre et al., "A Pretrainer's Guide to Training Data: Measuring the Effects of Data Age, Domain Coverage, Quality, & Toxicity".     arXiv:2305.13169 [cs.CL]

[10] Hubinger et al., "Risks from Learned Optimization in Advanced Machine Learning Systems". arXiv:1906.01820 [cs.AI] (2019).

[11] Qian et al., "ChatDev: Communicative Agents for Software Development". arXiv:2307.07924 [cs.SE] (2023).

---

### Author Response · Authors · 2024-08-12
**Overview of Contributions, Experiment Details, and Structural Improvements**

We thank the reviewers for their valuable feedback and insights provided on our submission. We are very happy to see a positive reception of our work and to be granted the opportunity to answer any questions that the reviewers might have.

We understand that there may have been some misunderstandings in interpreting our work, and as such, we would like to reiterate the core contributions of this work, explain our experimental setup, and describe our commitments to improving our writing in order to make it more accessible to readers.

## Key Contributions

* We are, to the best of our knowledge, the first to introduce the AI community to the topic of **secret collusion** in a comprehensive, formalized manner that rests on a strong theoretical and empirical foundation.
  * We reconcile epistemic concepts of knowledge with the contemporary AI and security literature to introduce a concise definition of decentralized systems of generative AI agents.
  * We present a thorough formalization of steganographic techniques, grounding, common knowledge, subliminal channels, and other directly relevant topics.
  * We identify and formalize the incentives required to perform secret collusion and understand the role of monitoring systems.
* We introduce a novel technical evaluation framework for quantifying secret collusion and introduce a number of benchmark scenarios, including individual model capability scenarios, group scenarios, and real-world inspired settings.
* We provide thorough empirical results for our framework and derive actionable insights about the implications on AI risks. These results extend to not only results from individual models, but also settings involving groups of models and real-world scenarios, including:
  * Individual steganographic capabilities are improving based on model size.
  * There already exist real-world settings where frontier models collude without being explicitly prompted to do so.
  * In group settings, as LLMs advance, the balance between colluding models (offense) and overseer models (defense) may shift in favor of the colluding models, leading to higher rates of undetected covertext steganography even when the overseer is of equal capability.
  * LLMs are currently good at finding zero-shot simple Schelling points for agreement but poor at complex Schelling points, k-level reasoning, self-sampling, logit simulation, and generating minimum entropy couplings. However, these capabilities are also improving, and we provide the first foundations to test them.
* We provide a comprehensive overview of possible mitigation actions and provide novel theoretical insights for a variety of foundational aspects regarding mitigations, including:
  * the ability of LLMs to produce pseudorandom numbers
  * the intractability of preventing secret collusion via blackbox parameter optimization
  * the difficulty of suppressing random oracles due to source free access to randomness
  * using grounding functions to detect steganography
  * paraphrasing as an insufficient solution to preventing steganography due to remaining semantic entropy and paraphrasing decreasing task performance in many tasks.
  * directly applicable techniques to prevent collusion such as filtering pre-training data, preventing access to certain tools, and in-context instructions.

---

> ### Author Response · Authors · 2024-08-12
> **Overview of Contributions, Experiment Details, and Structural Improvements (Part 2)**
>
> ## Clarifying our Experimental Setup
>
> ### CASE Framework
>
> CASE is our standardized model evaluation framework which enables experiments of three key classes:
>
> * **Individual model capability evaluations**: we, for the first time, provide a formal framework for measuring specific steganographic abilities.
> * **Group model capability evaluations** include LLMs communicating (and in certain cases, colluding) in the presence of monitoring systems.
> * **Novel qualitative and real-world-inspired evaluation settings**.
>
> ### Individual Capability Evaluations
>
> Appendix F breaks down the capabilities and information models require to perform covertext modification-based, information-theoretic, and complexity-theoretic steganography. We define clear metrics, use standardized evaluation techniques from SOTA literature, perform several rounds (often over 50 for each LLM tested), and present concise results in Table 1 and detailed results in the Appendix. Section 5 provides more specific information about the methods and metrics used in our experiments.
>
> ### Experimental Foundations
>
> Section 4 explains the setup of decentralized collaborative LLM agents and the presence of an overseer, studying the types of monitoring systems and incentives for collusion. Moreover, our group evaluation setting is explained in Appendix H.2 which we have committed to also describing more clearly in the main body. Our novel qualitative evaluations part of CASE allow for LLMs to be tested under certain real-world conditions that could lead to secret collusion (see Section 6 and Appendix Sections J and K).
>
> ### Mitigations
>
> We study mitigations and find key issues with paraphrasing, suppressing random oracles, and mitigating steganography with parameter optimization. We present a theoretical solution using grounding functions for detection and suggest near-term measures such as in-context instructions, filtering pre-training data, and preventing access to tools that allow steganography.
>
> ## Improving the Accessibility and Structure of our Camera Ready Version
>
> Once more, we thank all reviewers for their efforts in evaluating our paper and for their pointers relative to areas of improvements. We acknowledge the suggestions we have received regarding our writing/structure and would like to use this opportunity to commit to, if accepted, make the following adjustments in our camera-ready version.
>
> * In efforts of **making our work more accessible to readers from a wider range of backgrounds**, we will adapt the introduction and abstract to include a better and more simpler explanation of steganography and its importance, what is meant by collusion, and what our main contributions are.
> * We will further **provide a succinct and straightforward description of the CASE framework in the main body** to dispel any confusions that may arise from the current version of our paper. Most notably, we will explicitly describe CASE as an evaluation framework, comprising three separate streams: Individual Model Capability Evaluations, Group Evaluations, and Real-World/Qualitative Evaluations. We will refine the main body to highlight more specifically both the methods and metrics outlined in Section 5 (Capability Evaluations and Metrics) so as to reduce dependence on the Appendix as well as the configuration of our group evaluations described in Appendix H.2. We believe that these changes will allow for a better understanding of our experimental framework from the main body itself.
> * Aiming to address some feedback from reviewers, we will **prioritize augmenting the mitigations section in the main body to discuss practical limitations, computational costs, and real world implementation settings** that we discuss in our rebuttals in order to improve the readability and applicability of the insights we present in our mitigations section.

---

### Decision · Program_Chairs · 2024-09-25

**Decision:**

Accept (poster)

**Comment:**

The authors presented the problem of Secret Collusion in Large Language Models (LLMs) and propose a method for risk assessment at the current stage.
The authors pointed out that LLMs can potentially be used like a shared codebook, allowing secret information to be exchanged between LLMs without prior information sharing.
The authors demonstrated that when LLMs generate encryption schemes, these schemes might match between LLMs, effectively functioning as a shared codebook.
The authors proposed several evaluation methods to address these concerns and highlighted that GPT-4, in particular, shows high performance in this context.
Reviewers agree that Secret Collusion is an important issue that requires further research.
However, there were also concerns regarding the readability of the paper as follows
* The paper is targeted for audiences with proper backgrounds in security, which makes it challenging for other readers to understand.
* Most of the contents are deferred to the appendix, making it difficult to assess the validity of each statement in the main body.

By revisiting the structure of the paper and organizing the content between the main text and the appendix, the readability of the paper could be greatly improved, thereby allowing the importance of this work to be more widely recognized.